# Analysis of the Rolling Process of Alloy 6005 in a Three-High Skew Rolling Mill

**DOI:** 10.3390/ma13051114

**Published:** 2020-03-03

**Authors:** Teresa Bajor, Anna Kulakowska, Henryk Dyja

**Affiliations:** 1Faculty of Production Engineering and Materials Technology, Czestochowa University of Technology, 42-201 Czestochowa, Poland; 2Faculty of Science and Technology, Jan Dlugosz University in Czestochowa, 42-200 Czestochowa, Poland; 3Research Network Łukasiewicz Metal Forming Institute, 61-139 Poznan, Poland; henryk.dyja@inop.poznan.pl

**Keywords:** aluminum alloy 6005, physical modelling, skew rolling, three-high mill

## Abstract

This paper presents the results of numerical modelling of the rolling process of aluminum alloy bars in a three-high skew mill. The purpose of the examination was to determine the optimal rolling temperature for this alloy. The numerical examination for aluminum alloy 6005 (AlZn5.5MgCu) was performed using the Forge3^®^-2D Plane strain state commercial software. The rheological properties of the examined alloy were determined from uniaxial compression tests done using the metallurgical process simulation system Gleeble 3800. The numerical analysis of the process of rolling 6005 alloy bars in a three-high skew mill was conducted within the temperature range of 150–350 °C and at a deformation of 0.29.

## 1. Introduction

The unabated interest in lightweight metal alloys, which has maintained for several decades, is caused by many factors, including those of engineering, economic and environmental nature. A growing demand is observed for lightweight constructional materials, which reduce the weight of manufactured elements, thereby directly contributing to a reduction of energy consumption and a lowering of exhaust gas emission levels. In enhancing the applicability of constructional materials, materials with an ultra-fine grained structure are increasingly used. Their application in engineering is due to their high mechanical and plastic properties, compared with materials produced in conventional plastic working processes. For the manufacture of such materials, SPD (Severe Plastic Deformation) methods are used.

The main criterion for classifying a manufacturing process to SPD methods is its capability to obtain very large deformations (a true strain ε of above 4–6), while retaining the cohesion of the material being deformed. The basic SPD method is Equal Channel Angular Pressing (ECAP), proposed by Segal [1,2,3,4,5]. This method is the most popular for getting homogenized ultra-fine grained structures with high mechanical properties for most materials. Despite many advantages, this method requires the use of a very low strain rate and many steps to obtain the desired finished product. Another method of SPD is High Pressure Torsion (HPT) [6,7,8]. This method also allows high mechanical properties to be achieved, but problems with getting a homogeneous microstructure and the dimensions of the finished product do not make HPT usable in industrial conditions. Other methods are also currently found in the market, which do make it possible to obtain ultra-fine grained materials on a commercial scale. One of these methods is Accumulative Roll Bonding (ARB) [9,10]. The ultra-fine grained structure is obtained by multiple assembling and rolling of sheets.

The KOBO [11,12] method was developed at the Faculty of Non-Ferrous Metals of the AGH University of Science and Technology in Krakow by professors A. Korbel and W. Bochniak. This method also allows material deformation obtained by powder metallurgy.

Rolling in a three-high skew mill with large roll skewing angles was proposed in the last century [13,14]. During rolling in an oblique three-roll mill, a state of stress is created in the metal that is close to universal compression with large shear deformation. The rolled angles of the rolls led to increased technological plasticity by over 300% [15]. This method has found wide applications not only for rolling steel bars but also for bars made of non-ferrous metals. For example, titanium alloys were studied by Skripalenko et al. [16]. The study of aluminum bars from 2XXX alloys was conducted by Belov et al. [17]. A 7075 aluminum alloy rolled by a three-high skew mill was studied by [15]. Magnesium alloys were studied by Dobatkin et al. [18] and Gryc et al. [19].

The continued interest in aluminum alloy products has caused the implementation of increasingly new technologies of their plastic working processes, which requires the proper design of the process to obtain products with high mechanical properties at a relatively small weight. The selection parameters of plastic working processes are currently based on numerical analyses of many technological variants, whereby the costs of implementing and adapting new technologies to industrial conditions can be reduced [20,21].

By modifying the deformation scheme, it is possible to improve the mechanical properties of the finished product. Methods based on the process of rolling bars in a three-high skew mill combine different deformation schemes. Proposing a new technology of producing bars of a specific material with higher mechanical properties requires a detailed analysis of the plastic working conditions to be made. For plastic working processes, a basic feature that characterizes the plastic formability of material is the yield stress, *σ_p_* [22], which is the value of the stress necessary for initiating and continuing the plastic flow of metal under the conditions of a uniaxial stress state occurring during simple tension or compression. This value depends on the conditions of conducted deformation, such as specimen temperature (*T*); the value of true strain (*ε*); strain increase over time *ε* (*t*), and strain rate reduction prior to the process and utilizing the thermal effect of plastic deformation, which occurs during the rolling process.

The aim of the study was to analyze the process of rolling aluminum alloy bars in a three-high skew mill with a deformation of *ε* = 0.29 in a single rolling pass and, based on numerical examinations, to determine the best temperature of rolling the concerned alloy.

## 2. Materials and Methods 

The test material chosen for the tests was aluminum alloy 6005 (AlZn5.5MgCu) with chemical compositions as given in Table 1. The mechanical properties are given in Table 2.

The numerical simulation can be only correct when exact data of materials behavior at processing conditions are known. Physical simulation is needed and it must be executed on multi-purpose thermal-mechanical testing devices accurately reproducing the real industrial processing conditions. The true stress and strain state occurring when rolling bars in a three-high skew mill is not easy to represent. In the numerical modelling of this process, it is of key importance to accurately determine the variations in yield stress based on rheological tests [23]. The first stage of this study was to determine the variations in the rheological properties of analyzed aluminum alloy. For this purpose specimen compression tests were done using the Gleeble 3800 simulator (Dynamic System Inc., Poestenkill, NY, USA). The geometry of the compression tests and the thermal cycle simulation are shown in Figure 1.

The specimen for compressive testing was a cylinder with a diameter of 10 mm and length 12 mm (Figure 1a). During compression testing, the specimen heating rate was 2 °С/s (Figure 1b). The compression tests by the Gleeble 3800 simulator were carried out for the parameters shown in Table 3. All the tests were conducted in a vacuum to eliminate the specimen surface oxidation during testing at high temperatures. In each variant tested, the strain rate was a constant value.

The next stage of the study was to determine the mechanical properties of alloy 6005 by applying a tensile test on a ZWICK Z/100 testing machine (Zwick/Roell Group, Ulm, Germany). The tests were carried out at the parameters shown in Table 3, after which Kolmogorov’s yield criterion was determined. The specimen for these tests is shown in Figure 2.

The numerical examination of the three-high skew mill rolling process (ZAO “ISTOK ML”, Moscow, Russia) was carried out using the following initial parameters: feedstock temperature, 150 °C, 200 °C, 250 °C, 300 °C and 350 °C; roll rotational speed, 100 rpm; friction low–very high Tresca; friction coefficient, 0.4; friction factor, 0.8; thermal conductivity for the heat exchange between the band and the rolls, 20,000 W/m^2^ K, thermal conductivity, 167 W/m K; and coefficient of thermal expansion–linear, 23.4 µm/m K. A triangular finite element mesh was used to discretize the investigated volume of feedstock and tools.

The shape and dimensions of the working rolls, their positioning in space with indicated cross-sections I, II and III (respectively, at the entry to the deformation zone, in the middle of the deformation zone and at the exit from the deformation zone), in which stress and strain distributions and rolled band temperature were analyzed, are shown in Figure 3. The angle of inclination of the roll surface trajectories was 18°, while the angle of inclination of the roll generating line to the neutral axis of the rolling process in the deformation part was equal to 9°.

As the feedstock of the examined aluminum alloy was rolled, its diameter was reduced from 30 mm to 26 mm. The elongation factor in a rolling pass was λ = 1.33, while the true strain was equal to 0.29.

## 3. Results and Discussion

### 3.1. The Rheological Properties of Alloy 6005

Based on the plastometric tests, diagrams of the relationship of stress versus true strain for the investigated alloy were developed (Figure 4a–e) and the coefficients of the yield stress function were selected as per the Hensel–Spittel equation (Equation (1)) [25]. Relationship (1) was used to determine the value of *σ_p_* in the software during numerical modelling of the three-high skew mill rolling process. After the approximation of the plastometric test results, the coefficients of Equation (1) were determined. The values of these coefficients are given in Table 4.
(1)$$\sigma_{p}=Ae^{m_{1}T}\varepsilon^{m_{2}}{\dot{\varepsilon }}^{m_{3}}\varepsilon^{\frac{m_{4}}{\varepsilon }}{\left(1+\varepsilon \right)}^{m_{5}T}\varepsilon^{m_{7}\varepsilon }{\dot{\varepsilon }}^{m_{8}T}T^{m_{9}}$$
where: *σ_p_* is flow stress, MPa, *ɛ* is true strain, $\dot{\varepsilon }$ is strain rate, s^−1^, *T* is temperature, °C, *m*_1_–*m*_9_ are coefficients characterizing aluminum alloys.

From the analysis of the data represented in Figure 4a–e it can be found that, in the entire examined range of temperatures and strain rates, alloy 6005 was characterized by constant work hardening with the increase in deformation value. All of the obtained *σ_p_*-*ε* curves were monotonically ascending.

The analysis of the shape of the experimental and approximated work-hardening curve for alloy 6005 in the deformed specimen temperature range of 150–250°C showed that good agreement occurred between the true yield stress values and the values obtained from the approximation. With the increase in deformed specimen temperatures, the approximation error grew. Above the temperature 250 °C, structural changes occurred in the examined alloy (a hardening β″ phase occurred) [26]. The activation energy of the process of precipitation of hardening phases in aluminum alloys of group 6xxx depends on the contents of the main alloying elements. The value of the activation energy of β′′ phase precipitation in alloy 6005 is 66.6 kJ/mol, as has been shown in research [26]. The above-mentioned changes occurring in the material are an important element of the deformation process and have a major impact on the behavior of strain characteristics at high temperatures. The approximation method used did not allow for exothermic and endothermic processes occurring within the material, which had a decisive impact on the value of the approximation error (Table 4).

Examples of structure pictures taken after individual deformation stage are shown in Figure 5. Figure 5 shows a picture of the initial structure of the investigated alloy, where homogeneous grains of a similar size can be seen (a) and the alloy structure deformed at 0.1 s^−1^, at 350 °C, for real deformations *ε* = 0.8 (b).

The deformation resulted in the Al alloy grain refining in relation to the initial structure of the alloy and a slight grain elongation in the direction perpendicular to the applied compressive force was noticed (Figure 5b).

### 3.2. Kolmogorov’s Criterion

The three-high skew mill rolling process is a process in which a complex strain state occurs; therefore, it is justifiable to determine the plastic deformability of a given material without breaking its integrity using Kolmogorov’s criterion [27]. When plotting limiting plasticity graphs using Kolmogorov’s methodology, the limiting value of deformation Λ*_p_* is determined from a tensile test [28]:(2)$$\Lambda_{p}=2\sqrt{3}\ln\left(\frac{d_{0}}{d_{p}}\right)$$
where *d*_0_, *d_p_* are the initial specimen diameter and the specimen diameter at the breaking point, respectively.

The limiting deformation value can be expressed by the reduction of area *Z* [28]:(3)$$\Lambda_{p}=1.73\ln\left[100/\left(100-Z\right)\right]$$
where: $Z=\frac{S_{0}-S_{p}}{S_{0}}100\%$; *S*_0_, *S_p_* are the initial and end specimen cross-section surface areas at the breaking point, respectively.

The plasticity of aluminum alloy 6005 was assessed after determining the value of the relative reduction of area *Z* and the limiting value of deformation Λ*_p_*. The relationships of the relative reduction of the area and the limiting deformation value as a function of the test specimen temperature, *Z* = *f* (*T*) and Λ*_p_* = *f* (*T*), were then worked out (Figure 6a,b).

The relationships illustrated in Figure 6a show that both the magnitude of the relative reduction of the area and the limiting deformation value increased with the increase in feedstock temperature, except for the temperature range of 150–200 °C, where a slight drop in the relative reduction of the area and limiting deformation value was observed. The plasticity criterion according to Kolmogorow shows the limit value of deformation that can be set for a particular alloy. In the rolling process, deformation parameters should be used so as not to damage the material. The fact that the material was not cracked in the rolling process indicated that the rolling parameters were well selected. In the rolling process, the limiting deformation value was lower than that obtained in the alloy plasticity tests, which was due to the technical capabilities of the device (three-high skew mill).

Plastometric tests were conducted to determine the yield stress. Determining the yield stress value was necessary to perform numerical tests under specific deformation conditions in the three-high skew mill.

### 3.3. Numerical Modelling

The numerical analysis of the skew rolling process was made with Forge3^®^ commercial software (2.0NxT, Transvalor, Annapolis, France) using plastometric test results. Due to the axially-symmetrical character of the process, the computations were made for a plane state strain. This paper presents the analysis of the effect of the initial feedstock temperature on the distributions of stress intensities, strains in the roll gap, and the temperature in the cross-section of the rolled bar. The analysis was made for three planes, as shown in Figure 3. The first of the planes was situated at ^1^/_3_ of the deformation zone length, the second one at ^2^/_3_ of the deformation zone length, and the last one in the plane of the exit band from the deformation zone.

Figure 7 shows the distributions of Von Mises stress on the longitudinal sections in the middle band part for the feedstock at a temperature of 150–350 °C.

The obtained von Mises stress distributions showed that the greatest values, amounting to approx. 180 MPa, were obtained from rolled feedstock at a temperature of 150 °C (Figure 7a). With the increase in temperature, the value of stress intensity decreased to the value of 140 MPa, 120 MPa and 100 MPa, respectively, while the band rolled from the feedstock at 350°C attained the lowest value of 80 MPa (Figure 7e). This dependence of Von Mises stress on temperature was confirmed by plastometric tests.

Figure 8 shows the distributions of cumulated plastic strain on longitudinal sections and cross-sections in the middle part of bands rolled in a three-high skew mill for feedstock at a temperature of 150–350 °C.

From the data illustrated in Figure 8 it was noticed that the greatest cumulated plastic strain values occurred in the top layers being in contact with the rolls. For feedstock at a temperature of 150 °C, cumulated plastic strain was the greatest, amounting to 10, while for feedstock heated up to higher initial temperatures, plastic strain was lower.

With the increase in initial feedstock temperature, a reduction in the thickness of the intensively deformed layer was also observed.

In inner metal layers, the cumulated plastic strain value was much smaller, amounting to about three. This indicated that the material was worked there to a lesser degree, compared with its upper layer.

Temperature distributions on the longitudinal section in the middle part, and in selected cross-sections of the band rolled from the feedstock heated up to a temperature of 150–350 °C, are represented in Figure 9.

The analysis of the temperature distributions determined for longitudinal sections and cross-sections of deformed bars showed that an increase in metal temperature took place in the rolling process. In bars rolled from the feedstock at 150 °C (Figure 9a), an increase in temperature by approximately 100 °C was observed, while when rolling bars from the feedstock at 200 °C, the temperature increase was 80 °C (Figure 9b). For feedstocks at temperatures of 250, 300 and 350 °C (Figure 9c–e) the increase in metal temperature during the rolling process was 60, 40 and 25 °C, respectively.

The data illustrated in Figure 9 shows that the temperature distributions in the rolled bar cross-sections were uneven. A maximum temperature occurred in the surface zones of the bar, while a minimum temperature occurred in its axis.

From the data in Figure 9, it was noticed that the greatest gradient of temperature occurred in the center of the deformation zone, and then it decreased as the bar moved towards the plane of exits from the deformation zone. The greatest temperature gradient was observed in zone II (Figure 9), while the smallest gradient was observed at the beginning of the deformation zone (zone I).

During the numerical modelling of the bar skew rolling process, the variations in rolling power value for a single roll were also determined. Test results obtained for the process of rolling bars from the feedstock at a varying temperature are shown in Figure 10.

The data illustrated in Figure 10 shows that with the increase in the feedstock temperature from 150 °C to 350 °C, a significant reduction in the rolling power magnitude by approximately 55% resulted, which was a technologically advantageous phenomenon. A decrease in rolling power contributed to a lesser wear of the tools and a reduction of the energy intensity of the rolling process.

## 4. Conclusions

From the model studies carried out for alloy 6005, the following conclusions were drawn:-for initial specimen temperatures higher than 250 °C, the reserve of material plasticity increased;-the thermal effect was determined correctly, as confirmed by plastometric tests;-for temperatures above 250 °C, a more uniform stress distribution occurred on the rolled bar cross-section, which contributed to better mechanical properties of finished products;-the application of the process of rolling in the three-high skew mill enabled the use of work of deformation for initiating transformations to occur within the material without heating the feedstock up to a high temperature. For alloy 6005, the value of the activation energy of hardening β″ phase precipitation was attained at the initial deformation condition after the feedstock was heated up to a temperature of 150 °C, owing to a thermal effect that occurred at this temperature;-using numerical modelling methods it was possible to determine the optimal technological conditions of the process of manufacturing bars of aluminum alloy 6005 in a three-high skew mill, which enabled a finished product of high mechanical properties to be obtained.

## Figures and Tables

**Figure 1 materials-13-01114-f001:**
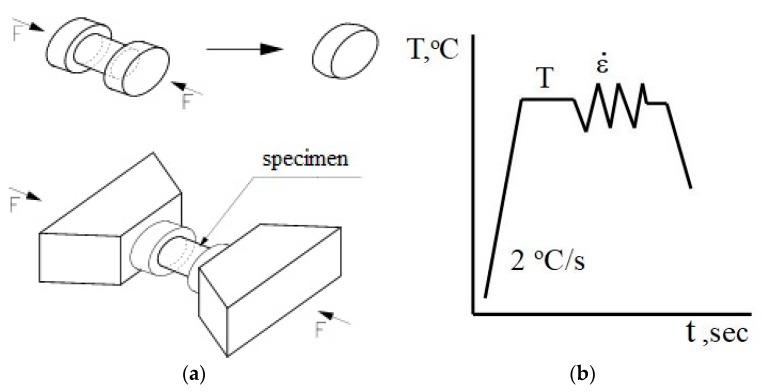
Geometry and dimension of load (**a**); and thermal cycle of physical simulation (**b**).

**Figure 2 materials-13-01114-f002:**
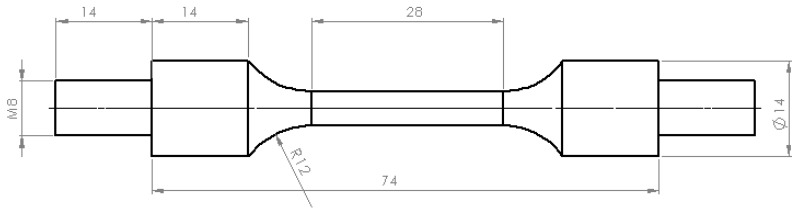
Geometry of specimen for tensile tests.

**Figure 3 materials-13-01114-f003:**
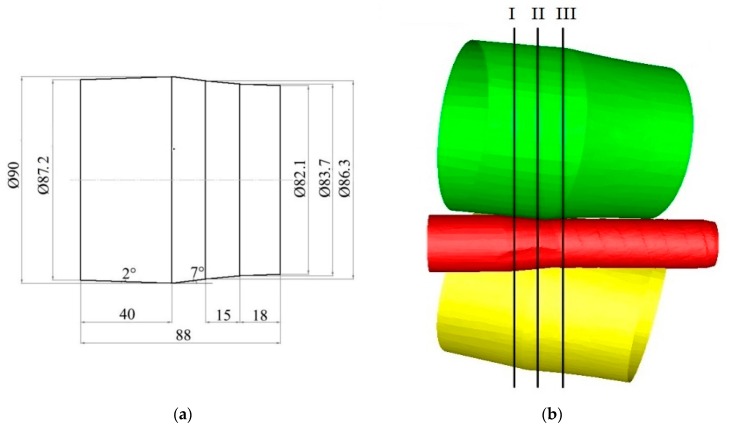
The shape and dimensions of the rolls (**a**) and the scheme of their arrangement in the rolling mill working space (**b**).

**Figure 4 materials-13-01114-f004:**
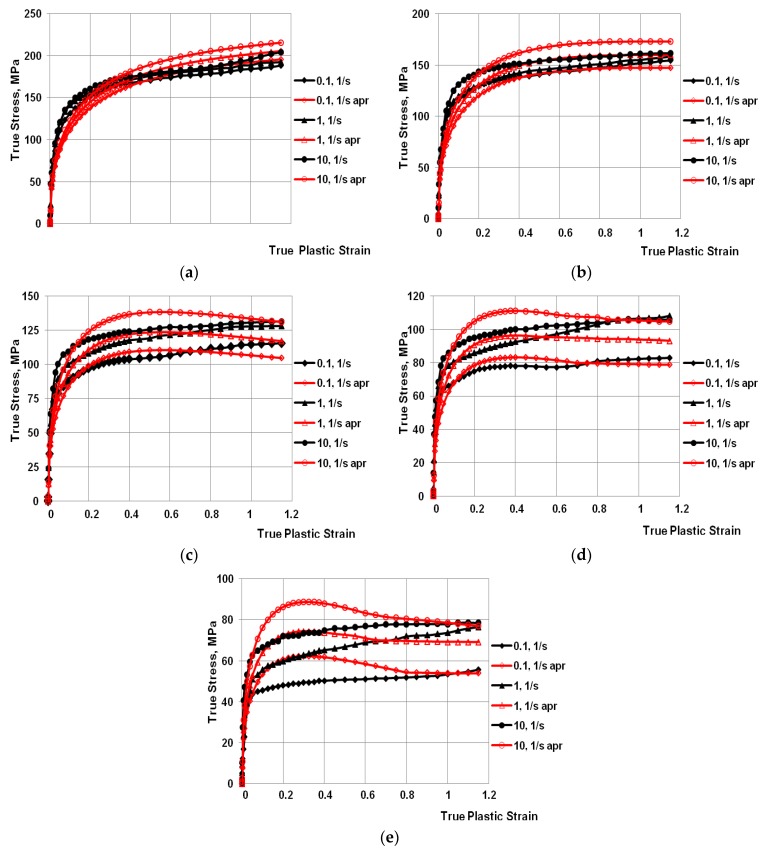
Work-hardening curves for the strain rate range of (0.1 s^−1^–10.0 s^−1^) at temperature of: (**a**) 150 °C; (**b**) 200 °C; (**c**) 250 °C; (**d**) 300 °C; and (**e**) 350 °C. (Black indicates the experimental curves; red indicates the approximated curves).

**Figure 5 materials-13-01114-f005:**
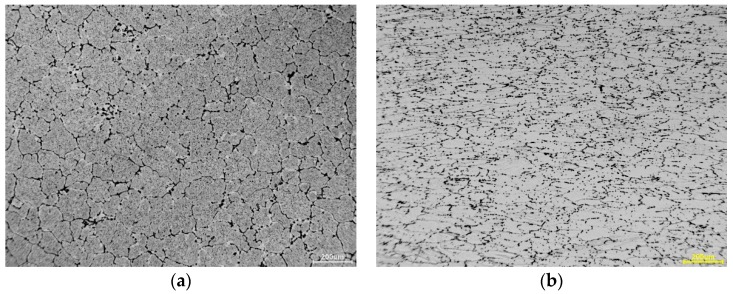
(**a**) The initial structure of the 6005 alloy, and (**b**) the structure of the alloy deformed at a strain rate of 0.1 s^−1^ at a temperature of 350 °C for deformation amounting to ε = 0.8.

**Figure 6 materials-13-01114-f006:**
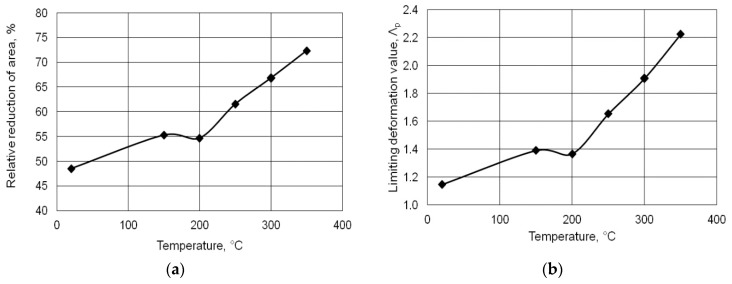
Relative reduction of area *Z* (**a**) and the limiting deformation value Λ*_p_* vs. temperature (**b**).

**Figure 7 materials-13-01114-f007:**
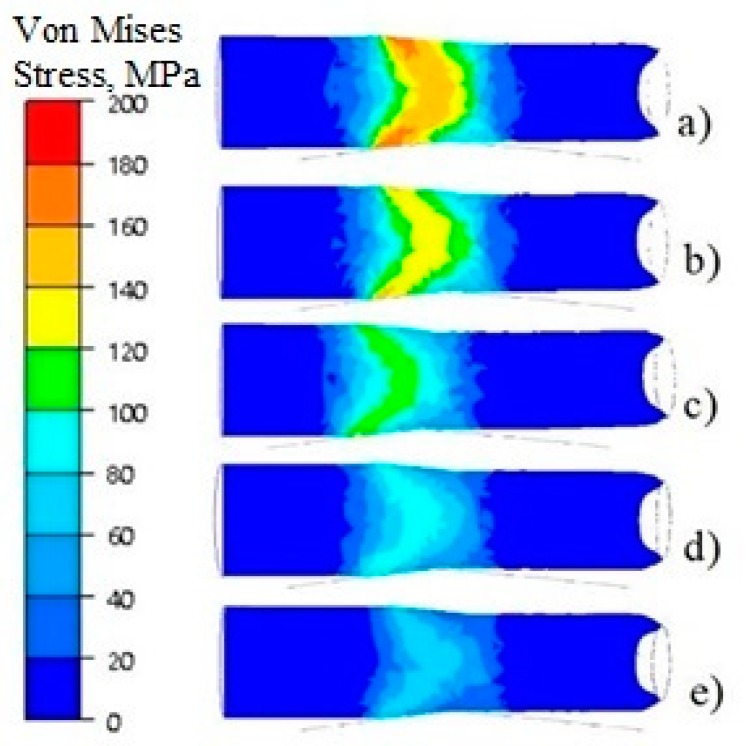
Von von Mises stress distributions on the longitudinal sections of deformed bars at a temperature of, respectively: (**a**) 150 °C; (**b**) 200 °C; (**c**) 250 °C; (**d**) 300 °C; and (**e**) 350 °C.

**Figure 8 materials-13-01114-f008:**
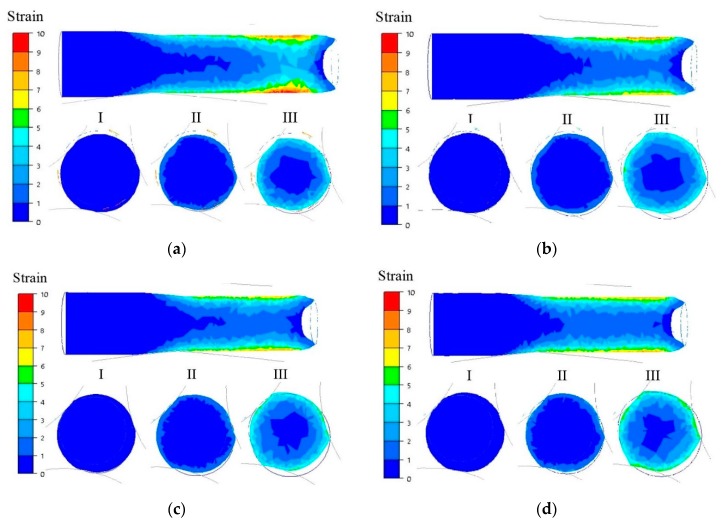
Distributions of cumulated plastic strain on the longitudinal sections and cross-sections of deformed bars at a temperature of, respectively (**a**) 150 °C; (**b**) 200 °C; (**c**) 250 °C; (**d**) 300 °C; and (**e**) 350 °C.

**Figure 9 materials-13-01114-f009:**
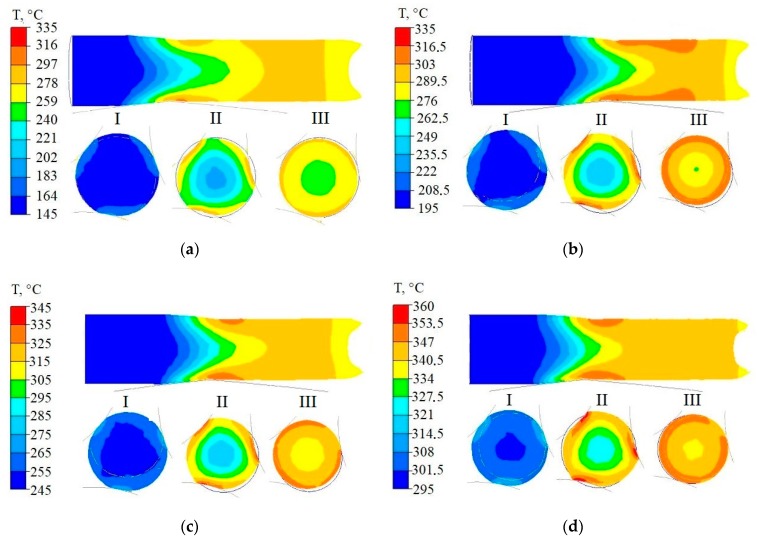
Distributions of temperature on the longitudinal sections and cross-sections of deformed bars at a temperature of, respectively, (**a**) 150 °C; (**b**) 200 °C; (**c**) 250 °C; (**d**) 300 °C; and (**e**) 350 °C.

**Figure 10 materials-13-01114-f010:**
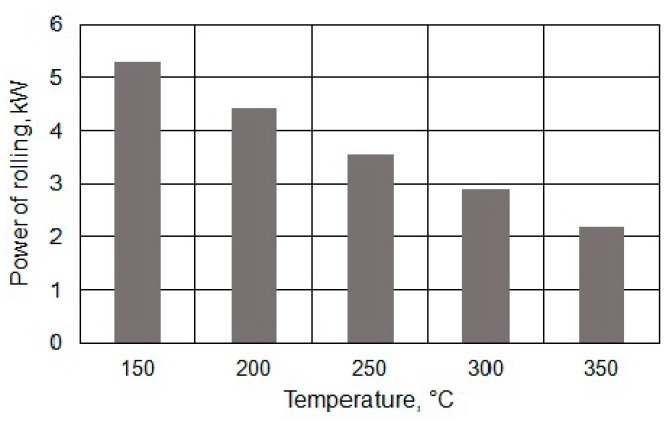
Variation in rolling power for a single roll in the bar rolling process.

**Table 1 materials-13-01114-t001:** Composition of the investigated alloy [%].

Alloy	Si	Fe	Cu	Mn	Mg	Cr	Zn	Ti	Al
6005	0.40	0.175	0.05	0.05	0.675	0.05	0.05	0.05	R

**Table 2 materials-13-01114-t002:** Mechanical properties of the investigated alloy.

*Rp*_0.2_, MPa	*R_m_*, MPa	*A*, %	*Z*, %
33.00	104.09	17.12	48.48

**Table 3 materials-13-01114-t003:** Parameters of compression and tensile tests.

Kind of Test	Temperature (*T*), °C	Strain Rate ($\dot{\varepsilon }$), s^−1^	Number of Specimen for Each Variant
Compression test	150, 200, 250, 300, 350	0.1, 1, 10	5
Tensile test	150, 200, 250, 300, 350	According to [24]	5

**Table 4 materials-13-01114-t004:** Values of the parameters A and *m*_1_–*m*_9_ used for determining the value of *σ_p_* for aluminum alloy 6005.

Aluminum	The Values of the Parameters Obtained from the Approximation of Equation (1)									*Average Approx. Error A [%]*
	*A*	*m* _1_	*m* _2_	*m* _3_	*m* _4_	*m* _5_	*m* _7_	*m* _8_	*m* _9_	
**6005**	6.304367	−0.00654	0.381576	−0.03346	−0.00017	−0.00560	0.265601	0.000317	0.963414	6.5

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
