# Peer review of "Analysis of the Rolling Process of Alloy 6005 in a Three-High Skew Rolling Mill"

_materials, 2020, doi:10.3390/ma13051114_

Round 1
Reviewer 1 Report
In this paper, the process of rolling Al alloy bars in a three-high skew mill was investigated through numerical modeling. Still, some modifications or clarifications should be made before the possible acceptance. 1. In page 1, line 17 and other place in the text. The sign “÷” was wrong. 2. In page 5, section 3.2. The parameter dp was the diameter at the breaking point in Kolmogorov’s methodology, while the alloy was not fracture in the process of rolling bars. The same as the parameter Sk. The authors should explain it. 3. How is the microstructure of the bar after plastometric test? 4. The relation between plastometric test and numerical results should be further analyzed.Author Response
Dear Reviewer,
We appreciate your valuable comments and detailed suggestions on our manuscript “Analysis of the process of rolling alloy 6005 in a three-high skew rolling mill”, which was submitted to MATERIALS Special Issue: “Development of the Rolling Process in Metallic Materials”

Reviewer 2 Report
The subject of the paper is interesting and important; numerical modelling and physical simulation are up-to-date methods, applying these methods is advantageous for both the knowing of the behaviour of different materials under different loading conditions and the developing of technological and production processes. Unfortunately, the presentation of the research work and the results is overall not unambiguous, significant elements and details have not been described. Consequently, the manuscript should be amended.
The literature review under the “1. Introduction” section is excessively compact, 18 references have been elaborated and 15 have been described or mentioned very shortly. However, relevant and important topics are missing, therefore the literature review should be partially detailed and new items should be added. Some suggestions for the amendment of the review are as follows.
(i) Mandziej, S. T.: Physical simulation of metallurgical processes. Materials and technology 44 (2010) 3, 105–119.
(ii) Lukács, J.; Kuzsella, L.; Koncsik, Zs.; Gáspár, M.; Meilinger, Á.: Role of the Physical Simulation for the Estimation of the Weldability of High Strength Steels and Aluminum Alloys. Materials Science Forum 812 pp. 149-154. , 6 p. (2015); DOI: 10.4028/www.scientific.net/MSF.812.149
(iii) Ramulu, P. J.: (November 27th 2019). Aluminum Alloys Behavior during Forming [Online First], IntechOpen, doi: 10.5772/intechopen.86077. Available from: https://www.intechopen.com/online-first/aluminum-alloys-behavior-during-forming.
Please, add supplementary information about the materials dependence of the referred methods (SPD, ECAP, HPT, ARB, KOBO, and three-high skew mill).
In the “2. Materials and Methods” section authors describe several important details, besides for the clarity of the research further information should be added. Please, add the mechanical properties of the investigated material under the delivery condition. Please, add the geometry and the dimensions of the used specimens, and the number of tested specimens for each case. Please, add a figure containing the thermal cycle during the physical simulation process. Please, explain the content of “the mode of strain increase in time ε (t)” (rows 56-57).
Further information should be added to “3. Results and Discussion” section. Please, consider and specify or refine your statement in rows 115-116: “good agreement occurs between the actual yield stress values and the values obtained from approximation”. Please, add characteristic micrographs to subsection 3.1, belonging to the structural change. Please, add further information belonging to your statement in rows 147-148: “which can be explained by changes occurring in the structure of the examined alloy at those temperatures”.
I have suggestions for the improving of the understandability of the manuscript, please:
- use dp and Sk consequently;
- consider the using of terms “stress intensity” and “strain intensity” and their applications in accordance with the relevant figures;
- use “plain strain state” instead of “plane state of strain” (row 152);
- correct the figure chapter in Figure 3 (row 143);
- correct the typing errors: “SDP” (row 31), “Figures 3a÷e” (row 98), “Figure 3a” (row 163), “Figure 3e” (row 166), “40 i 25” (row 191).
Although I am not English native speaking person, but I think that English language and style of the manuscript should be corrected.
Author Response
Dear Reviewer,
We appreciate your valuable comments and detailed suggestions on our manuscript “Analysis of the process of rolling alloy 6005 in a three-high skew rolling mill”, which was submitted to MATERIALS Special Issue: “Development of the Rolling Process in Metallic Materials”

Reviewer 3 Report
The scientific paper proposes a numerical analysis of 3D skew rolling mill process concerning a 6005 aluminium alloy starting from the identification of a mathematical model describing true stress plastic strain flow using compression tests (Gleeble 3800 plastometer and hydraulic Zwick machine . Optimal thermal and kinematics rolling conditions have been search based on goal o have a best homogeneity of plastic deformations and temperatures.
After a scientific literature review summarized in the Introduction paper part, based on previous research work interested on analysis of severe plastic deformations processes and analysis of rolling process, authors use a well structured content: Materials and Methods, Results and Discussions (The rheological properties of alloy 6005 – to define equation of the plastic flow, Kolmogorov's criterion – to define limiting value of plastic strain, Numerical modelling – to analyse results of finite element simulations using Forge3® in a plane strain state) and finally Conclusions.
The Conclusions identify and summarize the principal idea, but it is missing to well precise all the optimal rolling conditions obtained from the numerical analysis.
Furthermore a lot of major corrections are compulsory before the decision of publication by Journal Editors.
1° Page 1: Concerning the abstract it is necessary to replace theoretical analysis by numerical analysis and Forge3® by Forge3® - 2D plane strain state.
2° Page 2: it is compulsory to add a synthesis Table with all experimental conditions made by Gleeble3800 and Zwick Z/100 compression tests. It is necessary to specify “the use of a Coulomb-Tresca friction” and to specify all thermal properties of 6005 AA and of 3D rolling process (thermal convection and thermal conduction factor on material-tools interfaces).
3° The entire paper content must precise true plastic strain and true plastic strain-rate. So on Figure 2 it is required to define ordinates as the True Stress and Abscises as the True Plastic Strain.
4° Page 3: the authors must explain why are identified all the parameters of the used (necessary to be mentioned in the text the Hensel-Spittel equation as the name of equation (1) and to make the corresponding reference []). Some equations terms are similar with respect to the hardening or softening shape descrition and combination of all used terms can explain the important mismatches between the computed flow stress and the experimental one. Softening due to dynamic recristallization, especially of Beta phase, due to severe plastic deformation (large plastic strains), it is not already reproduced by the graphs of Figure 2. This one is justified by the use of many softening terms in equation (1) which lead to miscellaneous of the balance between the hardening and softening. So to have a softening tendency, generally m7 has negative values and m4 positive values. The identification is then wrong probably caused by the simultaneously identification of all parameters, or in general it is necessary to selection the most representative terms and only if the identification results can not be satisfactory, concerning the confrontations with the experimental data, to add or use the other terms. On the identification point of vie this numerical phenomena is due to the existing parameter correlations which characterise local minimum of cost function used for identification. It is also required to add in Table 2 the global relative error of least-squares used cost function expressed in terms of the true stresses during the identification procedure.
4° Page 4: The legend of Figures 2 concerning the plastic strain rate values is not clarified. It is not possible to understand what values of plastic strain rate characterise each curve and if theses one represent only the initial plastic strain rate (during the compression the plastic strain rate varies with the specimen height diminution).
5° Page 5: the authors must explain the interest to use the Kolmogorov limiting plastic strain value and eventually to make comparisons of this one with the numerical values of plastic strains obtained in the following part (average platic strains can be computed for all nodal values).
6° Page 6 – Page 8: “theoretical analysis” must be replaced by “numerical analysis” and “program” by “commercial software”. Type of used finite element mesh together with the used refinement can be indicated by authors. All boundaries conditions in term of tool kinematics, contact conditions and thermal properties as the material-tool interfaces must be detailed. Moreover the used numerical model by authors is a 2D plastic strain state one or the paper search to analyse a 3D rolling process. This hypothesis of 2D plastic strain can not be use for a skew rolling, only if the simulation is made in the plane section allowing the direction of the skew angle. So the numerical results are not relevant, especially for the goal of process conditions optimisation. Comparisons between 3D numerical simulations and the 2D plane strain numerical analysis can be making by authors to justify this point, or to indicate the plane section used for the 2D simulations.
7° Page 6 – Page 7: Figures 4 and Figure 5 speak about stress intensity and strain intensity. It is required to define this one; probably these ones concern the Von-Mises Stress and the Cumulated Plastic Strain as defined by the Forge 3® software..
8° Page 9: replace “technological parameters” by “technological operating conditions”.
9° Page 9: the Conclusions part must be well structured and indicate the final optimal rolling conditions obtained from this scientific study applied in industrial applications.
Regarding all above major and compulsory corrections and recommendations, the paper must be revised by authors before any decision of Editorial Board for publication in Materials.
Author Response

(The authors gave the same response as above.)

Reviewer 4 Report
Notes on the article of Teresa Bajor, Anna Kułakowska and Henryk Dyja “Analysis of the process of rolling alloy 6005 in a three-high skew rolling mill”
The paper reports results of modeling of the process of rolling aluminum alloy 6005 (AlZn5.5MgCu) using a three-high skew mill. The authors studied the theoretical behavior of Al 6005 alloy during the deformation in a three-high skew mill at the temperature of 150 ÷ 350 °C and at a deformation of 0.29. The authors also tried to determine the optimal rolling temperature for this alloy. The results of this article have the high importance for understanding of rolling process of the 6005 alloy. This is an interesting and well-written report, which should be published after minor revisions that are listed below:
1) In an Introduction section the authors should pay more attention to the skew rolling milling and radial shear rolling processes. For example, the authors should take note to the works of M.M. Skripalenko et al., who conducted the studies on titanium alloys [M.M. Skripalenko et al. Forming features and properties of titanium alloy billets after radial-shear rolling//Materials, 2019, 12(19), 3179; https://doi.org/10.3390/ma12193179], N. Belov et al., who conducted the studies on aluminum alloys [N. Belov et al. Phase composition and mechanical properties of Al–1.5%Cu–1.5%Mn–0.35%Zr(Fe,Si) wire alloy//Journal of Alloys and Compounds, 2019, 782, 735-746, https://doi.org/10.1016/j.jallcom.2018.12.240] and S.V. Dobatkin et al., who conducted the studies on magnesium alloys [S.V. Dobatkin et al. Grain refinement, texture, and mechanical properties of a magnesium alloy after radial-shear rolling//Journal of Alloys and Compounds, 2019, 774, 969-979, https://doi.org/10.1016/j.jallcom.2018.09.065] and other studies.
2) P. 1, line 14 and P. 2, line 64: It should be written “AlZn5.5MgCu“ instead of “AlZn5,5MgCu”.
Author Response

(The authors gave the same response as above.)

Round 2
Reviewer 1 Report
The authors revised the article systematically, but not modified in the manuscript.
Author Response
Dear reviewer, after a review, the manuscript has been supplemented with comments from the first review. A native speaker also reviewed the text.
Reviewer 2 Report
Thanks to the Authors for the valuable amendments, the added information and the corrections of more details; the modifications were precisely highlighted in the manuscript.
The literature review has been modified significantly, paragraphs have been edited newly, relevant and important details have been added.
The most important mechanical properties of the investigated alloy have been added (Table 1), but what does R0.2 mean? Is it Rp0.2 value? Important information belong to physical simulation have been added; please, create accordance between notation in Figure 1 and Table 3 (temperature, strain rate). The geometry and the dimensions of the specimens used for mechanical tests have been added; unfortunately, the drawing of the specimen is incorrect, important lines are missing (round bar specimen).
The information belonging to equation (1), which can be found in rows 130-131, are inaccurate: ε is not true stress (see row 32 too), ?̇ is not true plastic strain. These should be corrected. The legends of both axes in Figure 2 have been changed, it was important in favour of the understanding. I wold like to repeat my previous suggestion, please, add characteristic micrographs to subsection 3.1, belonging to the structural change.
I have suggestions for the improving of the understandability of the manuscript, please:
- add decimal point and dot digit under R0.2 value (Table 2), even though the digit is equal to zero (0);
- use “t” instead of “τ” in Figure 1 (b);
- emend the typing errors in row 92? “Fugure 1. Geometry and dimention …”
- emend the typing error in Table 3: “Accorging to [24]”;
- correct the errors in row 130-131 “coeffitients coefficients 131 characterising aluminium alloy..”;
- use the designation of the investigated alloy consequently (“6005” row 77, “Al 6005” row 101, “Al6005” row 133);
- use dp and Sk consequently, if you use dp, use Sp, or rather, if you use Sk use dk;
- use “Von Mises” instead of “Von-Mises” in row 190;
- use the same format and add the missing information under “References” section (23 and 25, 27 and 28).
Although I am not English native speaking person, but I think that English language and style of the manuscript should be corrected.
Author Response
Dear Rewiever, thank you for your comment. All you suggestions have been corrected and included in the manuscript.

Reviewer 3 Report
According to the previous review of the scientific the important required compulsory major corrections are not made by the authors. It is compulsory to make the following corrections:
1° Page 1: Concerning the abstract it is necessary to replace Forge3D by Forge3® - 2D plane strain state.
2° Page 2: it is compulsory to add in the synthesis Table 3 the experimental conditions corresponding to the Gleeble3800 compression tests and those corresponding to the Zwick Z/100 compression tests. It is necessary to specify “the use of a Coulomb-Tresca friction” which adds of the friction law definition. It is also compulsory to specify all thermal properties of 6005 AA 3D rolling process: thermal mass capacity, conductivity, thermal convection change coefficient and thermal conduction factor on material-tools interfaces).
3° Page 3: the authors must explain why are identified all the parameters of the used (necessary to be mentioned in the text the Hensel-Spittel equation as the name of equation (1) and to make the corresponding reference []). Some equations terms are similar with respect to the hardening or softening shape descrition and combination of all used terms can explain the important mismatches between the computed flow stress and the experimental one. Softening due to dynamic recristallization, especially of Beta phase, due to severe plastic deformation (large plastic strains), it is not already reproduced by the graphs of Figure 2. This one is justified by the use of all softening terms in equation (1) which lead to miscellaneous of the balance between the hardening and softening. So to have a softening tendency, generally m7 has negative values and m4 positive values. The identification is then wrong probably caused by the simultaneously identification of all parameters, or in general it is necessary the selection of most representative terms and only if the identification results can not be satisfactory, concerning the confrontations with the experimental data, to add or use the other terms. This one explains the divergence of numerical true stress – true plastic strain shape curve as compared to the experimental ones. On the identification point of vie this numerical phenomena is also due to the existing parameter correlations which characterise local minimum of cost function used for identification. It is then required to add in Table 2 the global relative error of least-squares used cost function expressed in terms of the true stresses obtained at the end of the identification procedure.
4° Page 4: The legend of Figures 2 concerning the plastic strain rate values is not clarified. It is not possible to understand what values of plastic strain rate characterise each curve and if theses one represent only the initial plastic strain rate (during the compression the plastic strain rate varies with the specimen height diminution). The authors not explain this point in the paper text and curves legends.
5° Page 5: the authors must explain into the paper text the interest to use the Kolmogorov limiting plastic strain value and to make comparisons of this one with the numerical values of plastic strains: average plastic strains can be computed for specific nodal values.
6° Page 6 – Page 8: Type of used finite element mesh together with the used refinement can be indicated by authors. All boundaries conditions in term of tool kinematics, contact conditions and thermal properties as the material-tool interfaces must be detailed. It is necessary to replace “cumulation plastic strain” by “cumulated plastic strain”. This correction must be made for entire paper content.
7° Page 6 – Page 7: Figures 4 and Figure 5 speak about stress intensity and strain intensity. It is required to define this one by “Von-Mises Stresses” and the “Cumulated Plastic Strain” as defined by the Forge 3® software..
8° Page 11 - Conclusions: replace “the technological conditions” by “optimal technological conditions”. It is required the mentioned and precise quantitatively here all these optimal technological conditions obtained from numerical analysis made by the proposed researches.
Regarding all above major and compulsory corrections and recommendations, the paper must be revised by authors before any decision of Editorial Board for publication in Materials.
Author Response
Dear Reviewers,
We appreciate your valuable comments and detailed suggestions on our manuscript “Analysis of the process of rolling alloy 6005 in a three-high skew rolling mill”, which was submitted to MATERIALS Special Issue: “Development of the Rolling Process in Metallic Materials”. Detailed answers are included in the attached file

Round 3
Reviewer 2 Report
Thanks the Authors again for the valuable amendments, the added information and the corrections of more details; the modifications were precisely highlighted in the third version of the manuscript. Unfortunately, there are open questions without exact and acceptable answers; therefore minor corrections should be carried out.
I would like to repeat my previous question: what does R0.2 mean in Table 2? Is it Rp0.2 value? The information belonging to physical simulations have been corrected, the data in Figure 1 and Table 3 are consentaneous. Figure 2 has been changed, the drawing of the round bar specimen is correct. I have to repeat my previous thought: the information belonging to equation (1), which can be found in rows 132-133, are inaccurate: ε is not true stress (see row 32, and rows 69-70, too), ?̇ is not true plastic strain. These should be corrected. Thanks the Authors for the added characteristic micrographs, the manuscript has gone to more complete. Several parts of the references have been modified; consequently the retrieval can be accomplished.
Author Response
Dear Reviewer,
We apologized for this situation. We agree with your comments. In fact R0.2 is Rp0.2, we have changed it in Table 3. Description of eqiuation (1) is also was corrected.
Kind regards,
Teresa Bajor